# Retrospective Analysis of the Correlation of MSI-h/dMMR Status and Response to Therapy for Endometrial Cancer: RAME Study, a Multicenter Experience

**DOI:** 10.3390/cancers15143639

**Published:** 2023-07-15

**Authors:** Valentina Tuninetti, Luca Pace, Eleonora Ghisoni, Virginia Quarà, Francesca Arezzo, Andrea Palicelli, Vincenzo Dario Mandato, Elena Geuna, Gennaro Cormio, Nicoletta Biglia, Lucia Borsotti, Silvia Gallo, Annamaria Ferrero, Elena Jacomuzzi, Luca Fuso, Jeremy Oscar Smith Pezua Sanjinez, Andrea Puppo, Andrea Caglio, Chiara Rognone, Margherita Turinetto, Giulia Scotto, Massimo Di Maio, Giorgio Valabrega

**Affiliations:** 1Department of Oncology, University of Turin, Medical Oncology, Ordine Mauriziano Hospital, 10128 Turin, Italy; virgi.quara@gmail.com (V.Q.); andrea.caglio@edu.unito.it (A.C.); chiara.rognone@unito.it (C.R.); massimo.dimaio@unito.it (M.D.M.); giorgio.valabrega@unito.it (G.V.); 2Obstetrics and Gynaecology Unit, Ordine Mauriziano Hospital, Department of Surgical Sciences, School of Medicine, University of Turin, 10124 Turin, Italy; luca.pace@unito.it (L.P.); nicoletta.biglia@unito.it (N.B.); annamaria.ferrero@unito.it (A.F.); elena.jacomuzzi@virgilio.it (E.J.); lfuso@mauriziano.it (L.F.); jeremyoscarmith.pezuasanjinez@unito.it (J.O.S.P.S.); 3Department of Oncology, Immuno-Oncology Service, University Hospital of Lausanne-CHUV, 1005 Lausanne, Switzerland; eleonora.ghisoni@chuv.ch; 4Department of Precision and Regenerative Medicine-DiMePRe-J, University of Bari ‘Aldo Moro’, 70121 Bari, Italy; francesca.arezzo@uniba.it; 5Pathology Unit, Azienda USL-IRCCS di Reggio Emilia, 42122 Reggio Emilia, Italy; andrea.palicelli@ausl.re.it; 6Unit of Obstetrics and Gynecology, Azienda USL-IRCCS di Reggio Emilia, 42122 Reggio Emilia, Italy; vincenzodario.mandato@ausl.re.it; 7Department of Medical Oncology, Candiolo Cancer Institute, FPO-IRCCS, 10060 Candiolo, Italy; elena.geuna@ircc.it; 8Gynecologic Oncology Unit, IRCCS Istituto Tumori “Giovanni Paolo II”, 70124 Bari, Italy; gennaro.cormio@uniba.it; 9Interdisciplinar Department of Medicine, University of Bari “Aldo Moro”, 70121 Bari, Italy; 10SC Direzione Sanitaria, Ordine Mauriziano Hospital, 10028 Turin, Italy; lborsotti@mauriziano.it (L.B.); sigallo@mauriziano.it (S.G.); 11Gyn-Obst Unit, S. Croce e Carle Hospital, 12100 Cuneo, Italy; puppo.a@ospedale.cuneo.it; 12Department of Oncology, University of Turin, 10124 Turin, Italy; margerita.turinetto@unito.it (M.T.); giulia.scotto@edu.unito.it (G.S.)

**Keywords:** endometrial cancer, microsatellite instability, mismatch repair deficiency, platinum-based chemotherapy

## Abstract

**Simple Summary:**

EC is the most common gynecological malignancy, and increased incidence and disease-related mortality have been observed in recent years. Data on the response to first-line carboplatin plus paclitaxel in EC is limited. The RAME study is a retrospective analysis aiming to assess response to chemotherapy in MSI-h/dMMR and MSI-l/pMMR EC patients. In patients receiving platinum-based chemotherapy in a first-line setting, PFS and OS were numerically longer in the MSI-l/pMMR population compared to MSI-h/dMMR patients.

**Abstract:**

Background: There is poor evidence regarding sensitivity to chemotherapy in endometrial cancer (EC) based on microsatellite instability (MSI)/mismatch repair (MMR) status. Methodology: The RAME study is a retrospective analysis aiming to assess response to chemotherapy in MSI-high (h)/deficient (d) MMR and MSI-low (l)/proficient (p) MMR EC patients. Primary endpoints were recurrence-free survival (RFS) for patients with localized disease and progression-free survival (PFS) and overall survival (OS) in patients with advanced/recurrent disease. Results: A total of 312 patients treated between 2010 and 2022 in four high-volume Multicenter Italian Trial in Ovarian cancer and gynecological malignancies (MITO) centers were selected. In total, 239 patients had endometrioid EC (76.6%), 151 had FIGO stage I at diagnosis (48.9%) and 71 were MSI-h/dMMR (22.8%). Median age was 65 (range 31–91) years. Among patients with localized disease, median RFS was 100.0 months (95% CI 59.4–140.7) for MSI-l/pMMR and 120.9 months (60.0–181.8) for MSI-h/dMMR (*p* = 0.39). Seventy-seven patients received first-line chemotherapy for advanced/recurrent disease. Patients with MSI-h/dMMR ECs had a significantly worse OS (*p* = 0.039). In patients receiving platinum-based chemotherapy, no statistically significant differences in PFS (*p* = 0.21) or OS (*p* = 0.057) were detected, although PFS and OS were numerically longer in the MSI-l/pMMR population. Conclusions: Patients with metastatic MSI-h/dMMR EC receiving first-line chemotherapy had a significantly worse OS.

## 1. Introduction

Endometrial cancer (EC) is the most common gynecological cancer, and increased incidence and mortality have been observed in recent years, especially in developed countries [1]. About 3 percent of women will be diagnosed with uterine cancer at some point during their lifetime [1]. Whereas early-stage EC is associated with an excellent 5-year relative survival rate (96%), this rate decreases to 18% in patients with advanced/metastatic disease [2,3]. The randomized phase III trial NRG 209, which compared the previous standard of care, TAP (paclitaxel doxorubicin cisplatin), with carboplatin paclitaxel showed no difference in terms of OS, but a more favorable safety profile for carboplatin paclitaxel. This trial led to a new standard of care [4], with an overall response rate (ORR) of 50–60%, and a median progression-free survival (PFS) and an overall survival (OS) of 8 months and about 20 months, respectively [4,5,6].

Historically, EC was classified according to the Bokhman classification [7], as type I (mainly endometrioid histotype, often associated with obesity, hyperlipidemia, hyperglycemia and increased estrogens concentration, positive hormone expression with a good prognosis) or type II (usually non-endometrioid histotype, associated with a negative hormone expression, p53 mutated with a poor prognosis). In 2013, The Cancer Genome Atlas (TGCA) [8,9,10,11] introduced a new classification based on molecular analysis: EC is no longer considered as a single entity, but the set of types of different tumors, each with a completely different prognosis. The new classification includes four subgroups: DNA polymerase epsilon (POLE)-mutated tumors; tumors with high microsatellite instability (MSI) or DNA mismatch repair mechanism (dMMR) deficiency, which identifies patients at risk for Lynch syndrome (LS) [12,13]; tumors with p53 alterations (p53-mutants), or high copy-number; and tumors with no specific molecular profile (NSMP), or low copy-number. 

The first two are distinguished by a defect in DNA repair, a high mutational rate, and a high neoantigen load. These two subgroups are also inflamed due to an abundance of tumor-infiltrating lymphocytes and increased expression of programmed cell death protein 1 (PD-1) and PD-L1 as well as potentially more responsive to immune checkpoint inhibitors (ICIs) [14,15,16]. MMR is a highly conserved mechanism that restores DNA integrity by correcting single-base mismatches and insertion–deletion loops that can occur during DNA replication [17]. 

The last few years have witnessed the approvals of ICIs in the setting of advanced/metastatic EC previously treated with platinum-based chemotherapy: dostarlimab on the basis of the GARNET study [16] for MSI-h/dMMR, pembrolizumab on the basis of KEYNOTE-158 [18], again for MSI-h/dMMR, and pembrolizumab plus lenvatinib on the basis of the KEYNOTE-775 trial [19] for MSI-l/pMMR. The pembrolizumab-lenvatinib combination has also been tested in first-line treatment of advanced/metastastic endometrial cancers (LEAP 001), and the process of obtaining the results is ongoing. Recent research into the administration of ICIs in combination with chemotherapy has yielded very promising results. The ENGOT-EN6-NSGO/GOG-3031/RUBY trial [20] has provided strong data in terms of PFS and OS for the use of dostarlimab in combination with chemotherapy; furthermore, the NRG-GY018 study [21] demonstrated that combining pembrolizumab with standard chemotherapy resulted in significantly longer PFS than chemotherapy alone in both the MSI-h/dMMR and MSI-l/dMMR cohorts. A similar randomized trial comparing carboplatin paclitaxel with carboplatin paclitaxel atezolizumab is currently ongoing. As a result, the combination of ICIs and carboplatin paclitaxel will soon become a new standard for the treatment of advanced/metastatic EC. It is also important to highlight two randomized phase III trials in dMMR/MSI-h advanced/metastatic endometrial cancer that compare first-line chemotherapy with carboplatin paclitaxel with the same combination plus pembrolizumab (KEYNOTE C93 trial) or dostarlimab (DOMENICA). Both trials include crossover at progression and have OS as the primary endpoint. 

It is now well established that immunotherapy in MSI-h/dMMR EC is the best option both in first-line treatment in combination with chemotherapy and after the failure of first-line treatment with carboplatin and paclitaxel, and data from the recently completed PORTEC3 trial [22,23,24,25] suggested that adjuvant platinum-based chemotherapy had no benefit in MSI-h/dMMR EC. PORTEC-3 was an open-label, international, randomized, phase III trial that included patients with high-risk EC with FIGO 2009 stage I, endometrioid-type grade 3 with deep myometrial invasion or lymph–vascular space invasion (or both), endometrioid-type stage II or III, or stage I to III with serous or clear cell histology. Six hundred and sixty eligible patients were included in the final analysis, of whom 330 were assigned to chemoradiotherapy and 330 were assigned to radiotherapy. Patients were assigned to receive radiotherapy alone (48·6 Gy in 1·8 Gy fractions given on 5 days per week) or radiotherapy and chemotherapy (consisting of two cycles of cisplatin 50 mg/m^2^ given during radiotherapy, followed by four cycles of carboplatin AUC5 and paclitaxel 175 mg/m^2^). The co-primary endpoints were OS and failure-free survival (FFS). Adjuvant chemotherapy given during and after radiotherapy for high-risk endometrial cancer did not improve 5-year OS, although it did increase FFS. Women with high-risk EC should be individually counseled about this combined treatment.

However, limited data on the response to first-line carboplatin plus paclitaxel in EC are available [4].

We conducted a multicenter retrospective study with the aim of describing (i) RFS for patients with EC eligible at diagnosis for potentially curative treatment and treatment with surgery, and (ii) PFS and OS in patients with advanced/recurrent disease, in particular in patients treated with platinum-based chemotherapy.

## 2. Methods

Patients with EC with known MSI/MMR status treated consecutively between January 2010 and January 2022 in four high-volume MITO centers (Mauriziano Hospital, Candiolo Hospital, Bari Hospital and Reggio Emilia Hospital) were identified via a multicenter retrospective review of electronic case records. Eligibility criteria included adult patients with EC with any histology and stage of disease. Standardized chart review was performed, collecting date of diagnosis, age at diagnosis, comorbidities, date of initial local therapy, adjuvant treatment, MMR/MSI status, date of first relapse, subsequent treatments and types. Data cutoff was 30 April 2023. The study was approved by a central ethics committee, and informed consent was obtained from patients. A flowchart of the study is presented in Figure 1.

The primary endpoints of the study were RFS for patients with localized disease at diagnosis and treated with surgery, and PFS and OS in patients with advanced/recurrent disease. To this end, PFS and OS were calculated in the subgroup of patients receiving platinum-based chemotherapy. Response was assessed locally by the clinician, according to evaluation using a computed tomography (CT) scanner or positron emission tomography CT (PET-CT). RFS was defined as the time between date of diagnosis and first recurrence. PFS was defined as the time from the date of start of first-line therapy to disease progression or death. OS was defined as the time from the date of start of first-line therapy to death.

The MMR/MSI status was assessed locally, according to one of the standard practices. Methods for determination included loss of expression of 1 or more MMR proteins by immunochemistry (IHC), or instability in 2 or more of five tumor repeat loci by polymerase chain reaction (PCR) assay. When IHC was used, all four MMR proteins were assessed (MLH1, PMS2, MSH2, MSH6). In our series, 75% of the patients were tested with IHC, 8% with PCR and 17% with both the methods. When performing both methods of testing, results were concordant.

## 3. Statistical Analysis

Patients’ and tumor characteristics (age at diagnosis, comorbidities, histology, FIGO stage, surgery, MMR/MSI status and subsequent systemic treatment) were described (median for continuous variables and frequency for categorical variables). Median follow-up was estimated using the reverse Kaplan–Meier method. RFS, PFS and OS were estimated using the Kaplan–Meier method, and median was reported along with 95% CI. The study cutoff date for the statistical analysis was 30th April, 2023. Statistical analysis was performed by using IBM SPSS Statistics, version 28.0.1.0

## 4. Results

Between 1 January 2010 and 1 January 2022, we consecutively identified 317 patients treated for EC. The efficacy analysis was performed on 312 patients because of missing data regarding the MMR/MSI status of five patients. All the patients’ characteristics are summarized in Table 1.

Median age was 65.3 (31.5–90.9) years, with no differences between the two groups analyzed (MSH-l/pMMR vs. MSH-h/dMMR, *p* = 0.26). The majority of patients were MSH-l/pMMR (77.2%, 241/312), mainly detected with immunohistochemistry (92%). The details of MMR status are reported in Appendix A.

Most patients (71.2%, 22/312) had comorbidities: diabetes, hypertension and/or cardiopathy were the most frequent, but with no differences between the two groups. Two hundred and thirty-nine patients (76.6%) had endometrioid EC. Among the members of the MSH-l/pMMR group, patients with endometrioid EC accounted for 90.1% (*p* = 0.002). At diagnosis, nearly half of patients (48.9%) were FIGO stage I.

Among the 278 patients with no metastatic disease at diagnosis and treated with surgery, 171/278 (61.5%) underwent adjuvant therapy: 104/278 (37.4%) received radiotherapy, 96/278 (34.5%) received brachytherapy, 115/278 (41.4%) received chemotherapy and 6/278 (1.8%) received hormonal therapy. No differences between the two groups were highlighted according to the different adjuvant treatments (respectively *p* = 0.43 for RT, *p* = 0.34 for BT, *p* = 0.71 for chemotherapy, *p* = 0.21 for HT). At a median FU of 58.2 months, the median RFS was 100.0 months (95% CI 59.4–140.7) for MSI-l/pMMR and 120.9 months (60.0–181.8) for MSI-h/dMMR 0.81, 95% CI 0.50–1.31, *p* = 0.39). See Table 2 and Figure 2 for details. 

Seventy-seven patients received first-line CT for advanced/recurrent disease, 76.6% (59/77) received platinum-based CT and 19.5% (15/77) were MSI-l/pMMR. Median age was 68.3 (39.8–87.2) years, with no differences being observed between MSH-l/pMMR and MSH-h/dMMR (*p* = 0.72). 

Median FU was 22.5 months. In this setting, median PFS was 10.3 months (95% CI 7.7–12.8) in MSI-l/pMMR and 6.3 months (95% CI 2.0–10.6) in MSI-l/pMMR (HR1.53, 95% CI 0.79–2.97, *p* = 0.21). Median OS was 37.2 months (95% CI 28.0–46.4) for MSI-l/pMMR and median OS was 14.0 months (95% CI 1.0–27.1) for MSI-h/dMMR, with a significantly worse OS in MSI-h/dMMR patients (HR 2.26, 95% CI 1.04–4.92, *p* = 0.039) (Table 3 and Figure 3A). 

In the subgroup of patients receiving platinum-based CT, no statistically significant differences in PFS (*p* = 0.21) or OS (*p* = 0.057) were detected, but PFS and OS were numerically longer in the MSI-l/pMMR population (Figure 3B).

## 5. Discussion

In this study, we described a large series of 278 ECs treated for localized disease with surgery (±adjuvant treatments) and 77 advanced/metastatic ECs treated with first-line chemotherapy, mainly platinum-based chemotherapy. 

In accordance with the literature, our series consists of about 20% of MSI-h/dMMR EC [24,26]. More than 90% of MSI-h/dMMR EC have an endometroid histology. In the real-world setting of this study, the median age for advanced/metastatic EC was 68 years, which is slightly higher than that in major clinical trials investigating platinum-based chemotherapy in this setting (62 years in the Sovak et al. trial [27] and 65 years in the Pectasides D at al. trial [28]).

Currently, we know from the two most important first-line randomized trials (RUBY [20] and NRGY-018 [21]) that the addition of an ICI to carboplatin plus paclitaxel in advanced/metastatic EC improves PFS in both the dMMR population (HR 0.28 and 0.30 respectively), and the pMMR population (HR 0.64 and 0.54, respectively), with a better PFS in the control arm of the pMMR population. The RUBY study also demonstrated a statistically significant trend in OS in the overall population, with an increased benefit in quality of life (QoL) in the dostarlimab arm in patients with MSI-h/dMMR EC. 

In our retrospective real-world population of relapsed patients, which is similar to 50% of the RUBY population, we confirm a worse performance of carboplatin plus paclitaxel in relapsed dMMR EC (PFS 6 months) compared to the pMMR population (PFS 12 months). When considering the adjuvant setting, there is only preliminary evidence suggesting that MMR/MSI status in EC can also predict benefits from chemotherapy in the adjuvant setting [29]. 

The results of the PORTEC3 trial [22] confirmed a modest but statistically significant benefit from adjuvant chemotherapy in terms of RFS and OS in all comers, but suggested that adjuvant platinum-based chemotherapy offered limited benefit in MSI-h/dMMR EC. 

A post hoc analysis was performed to evaluate these results according to molecular classification: the p53-mutated tumors in immunochemistry presented an absolute benefit of adjuvant chemotherapy of 25% in terms of RFS, whereas there was no benefit for dMMR tumors [24,30]. 

The RAINBO umbrella program, which aims to identify the best adjuvant treatment in EC, is currently ongoing. Eligible patients will be assigned to one of the four RAINBOW trials (ClinicalTrials.gov Identifier: NCT05255653) based on the molecular profile of their cancers: p53 abnormal EC patients to the p53abn-RED trial; MSI-h/dMMR EC patients to the MMRd-GREEN trial; no specific molecular profile EC patients to NSMP-ORANGE trial and POLE mutant EC patients to the POLEmut-BLUE trial. Results could clarify the role of adjuvant treatments in a prospective setting. 

In our cohort of patients with localized disease, median RFS was 120 months in MSI-h/dMMR versus 100 months in MSI-l/pMMR with no clear statistically significant difference between the two populations [27,28]. 

None of the patients in this retrospective study had received ICIs, and therefore the data for response to chemotherapy were not influenced by previous exposure to immunotherapy. 

Our data suggested a numerically better RFS in MSI-h/dMMR (although the difference was not statistically significant), but when those patients experienced recurrence or in MSI-h/dMMR de novo EC, prognosis was worse in terms of both PFS and OS. 

Our retrospective study did not define a preplanned statistical power for either RFS analysis in patients with localized stage or for PFS and OS analyses in advanced disease. 

The analysis was limited to describing the prognostic differences between the groups analyzed, and we cannot comment definitively on the predictive role of MMR/MSI status, since all the patients enrolled in our study were treated in clinical practice, and no control group was employed.

Although the present study suffers from all of the limitations associated with its retrospective nature, our real-world data confirm that MSI-h/dMMR ECs show a modest outcome when treated with chemotherapy in the relapse setting. Conversely, pMMR patients seem to derive a greater benefit from carboplatin plus paclitaxel. 

## 6. Conclusions

In conclusion, although RFS is better in MSI-h/dMMR, as expected, since patients were not treated with ICIs, this does translate into an advantage in terms of OS. Our study therefore confirms the need to incorporate ICIs in the MSI-h/dMMR EC population, in line with the data from randomized phase III clinical trials. The optimal choice for relapsed pMMR EC patients remains to be defined, especially in patients with long RFS, where carboplatin paclitaxel may still be a treatment choice. 

## Figures and Tables

**Figure 1 cancers-15-03639-f001:**
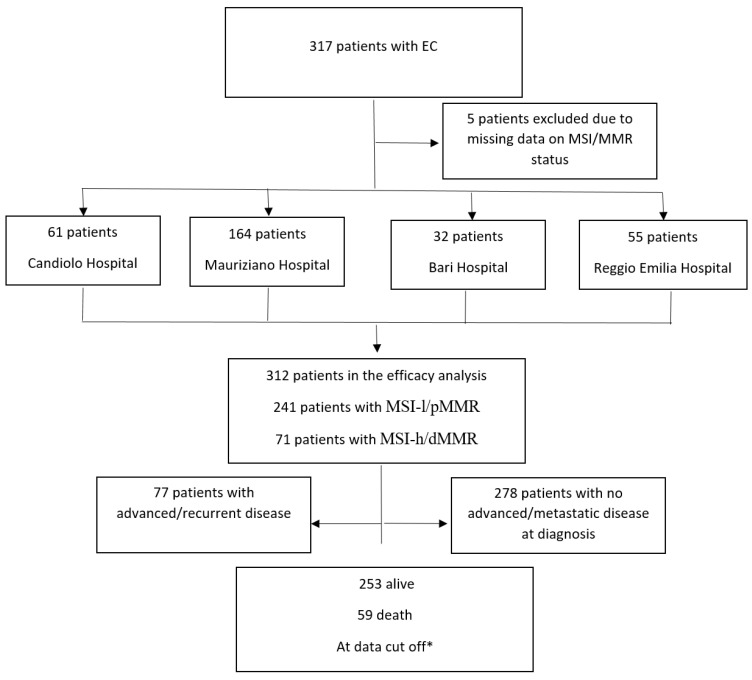
Flowchart of the study. * Data cutoff 30 April 2023.

**Figure 2 cancers-15-03639-f002:**
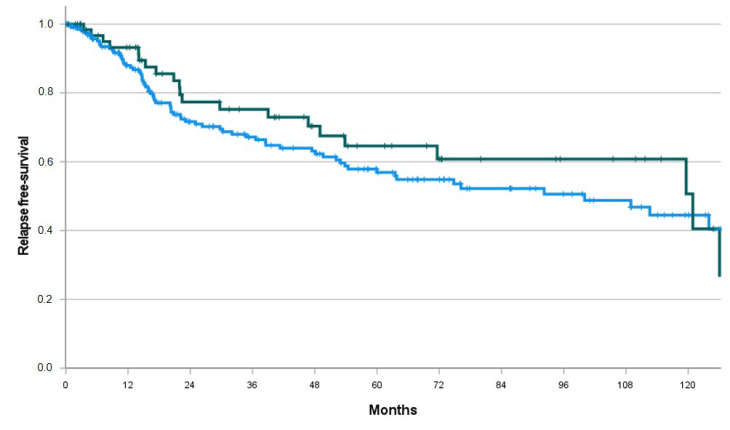
RFS by MSI/MMR status. Blue line, MSI-l/pMMR; green line, MSI-h/dMMR.

**Figure 3 cancers-15-03639-f003:**
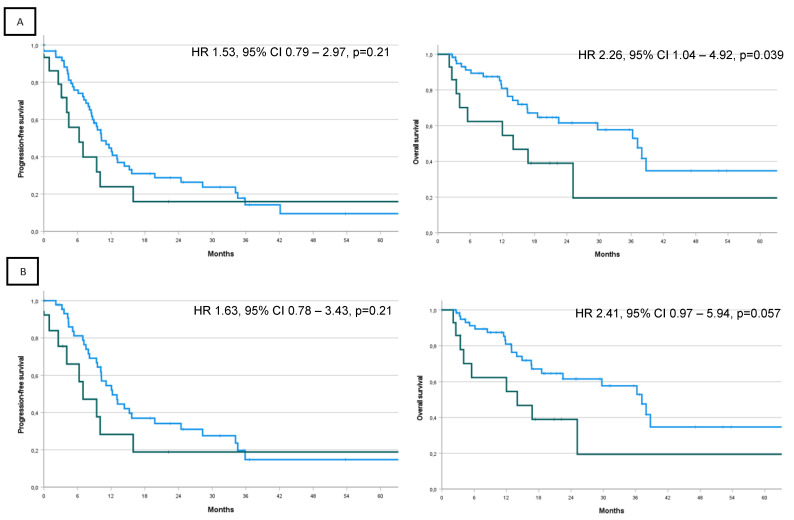
(**A**) PFS and OS by MSI/MMR status. (**B**) PFS and OS by MSI/MMR status in patient receiving platinum-based chemotherapy as first-line of treatment. Blue line, MSI-l/pMMR; green line MSI-h/dMMR.

**Table 1 cancers-15-03639-t001:** Patients’ characteristics at diagnosis.

	All Patients	MSI-l/pMMR Patients	MSI-h/dMMR Patients	*p*-Value
Center (No. of patients)	312	241/312 (77.2%)	71/312 (22.8%)	
-Mauriziano	164 (52.6%)	132 (54.8%)	32 (45.1%)	
-Candiolo	61 (19.6%)	44 (18.3%)	17 (23.9%)	
-Reggio Emilia	55 (17.6%)	40 (16.6%)	15 (21.1%)	
-Bari	32 (10.3)	25 (10.4%)	7 (9.9%)	
Median age at diagnosis (years, CI)	65.3 (31.5–90.9)	65.3 (31.5–90.9)	64.6 (42.4–89.8)	*p* = 0.26
Comorbidities				*p* = 0.66
-Yes	222 (71.2%)	170 (70.5%)	52(73.1%)	
-No	90(28.8%)	71(29.5%)	19(26.8%)	
Diabetes				*p* = 0.57
-Yes	42 (13.5%)	31 (12.9%)	11 (15.5%)	
-No	270 (86.5%)	210 (87.1%)	60 (84.5%)	
Hypertension				*p* = 0.73
-Yes	135 (43.3%)	103 (42.7%)	32 (45.1%)	
-No	177 (56.7%)	138 (57.3%)	39 (54.9%)	
Cardiopathy				*p* = 0.95
Yes	27 (8.7%)	21 (8.7%)	6 (8.5%)	
No	285 (91.3%)	220 (91.3%)	65(91.5%)	
Hystology				***p* = 0.002**
Endometrioid	239 (76.6%)	175 (72.6%)	64 (90.1%)	
Other	73 (23.4%)	66 (27.4%)	7 (9.9%)	
FIGO stage at diagnosis				*p* = 0.43
I	151 (48.4%)	115 (47.7%)	36 (50.7%)	
II	32 (10.3%)	23 (9.5%)	9 (12.7%)	
III	93(29.8%)	73 (30.3%)	20 (28.2%)	
IV	33 (10.6%)	28 (11.6%)	5 (7.0%)	
Missing data	3 (0.9%)	2 (0.8%)	1 (1.4%)	

No = number; MSI-h = high microsatellite instability; dMMR = mismatch repair deficiency; MSI-l = low microsatellite instability; pMMR = mismatch repair proficient; FIGO = International Federation of Gynecology and Obstetrics; CI = confidence interval.

**Table 2 cancers-15-03639-t002:** Characteristics of the 278 patients included in the analysis of RFS (data cutoff 30 April 2023).

	Whole Series (N = 278)	MSI-l/pMMR Patients (N = 212)	MSI-h/dMMR Patients (N = 66)	*p* Value
No. of patients with event (recurrence or death)	94/278 (33.8%)	73/212 (34.4%)	21/66 (31.8%)	
Adjuvant Therapy				*p* = 0.20
Yes	171/278 (61.5%)	126/212 (54.9%)	45/66 (68.2%)	
No	107/278 (38.5%)	86/212 (40.6%)	21/66 (31.8%)	
Radiotherapy				*p* = 0.34
Yes	104/278 (37.4%)	76/212 (35.8%)	28/66 (42.4%)	
No	174/278 (62.6%)	136/212 (64.2%)	38/66 (57.6%)	
Brachitherapy				*p* = 0.34
Yes	96/278 (34.5%)	70/212 (33.0%)	26/66 (39.4%)	
No	182/278 (65.5%)	142/212 (67.0%)	40/66 (60.6%)	
Chemotherapy				*p* = 0.71
Yes	115/278 (41.4%)	89/212 (42.0%)	26/66 (39.4%)	
No	163/278 (58.6%)	123/212 (58.0%)	40/66 (60.6%)	
Hormonotherapy				*p* = 0.21
Yes	5/278 (1.8%)	5/212 (2.4%)	0/66 (0%)	
No	273/278 (98.2%)	207/212 (97.6%)	66/66 (100%)	
RFS Median (months, CI)	112.6 (78.4–146.8)	100.0 (59.4–140.7)	120.9 (60.0–181.8)	*p* = 0.39
Rate at 1 year	89.3%	88.0%	93.2%	
Rate at 2 years	73.1%	71.7%	77.4%	
Rate at 3 years	69.2%	67.2%	75.2%	
Rate at 5 years	58.8%	56.9%	64.6%	
Rate at 10 years	46.0%	44.5%	50.7%	

No = number; RFS = recurrence free survival; CI = confidence interval. RFS was 83.9% at 1 year, 73.1% at 2 years, 69.2% at 3 years, 58.8% at 5 years and 46% at 10 years.

**Table 3 cancers-15-03639-t003:** Characteristics of patients (N = 77) treated with chemotherapy as first-line treatment for advanced/metastatic disease. N = number; PFS = progression free survival; OS = overall survival; CI = confidence interval; CT = chemotherapy.

	All Comers	MSI-l/pMMR Patients	MSI-h/dMMR Patients	*p* Value
No. of patients	77	62 (80.5%)	15 (19.5%)	
Median age at advanced/metastatic disease (years, CI)	68.3 (39.8–87.3)	67.9 (39.8–86.0)	71.0 (56.8–87.2)	*p* = 0.72
First-line CT				
platinum-based	59 (76.6%)	46 (74.2%)	13 (86.7%)	
liposomal doxorubicin	12/77 (15.6%)	11/62 (17.4%)	1/15 (6.7%)	
other	6/77 (6.8%)	5/62 (8.1%)	1/15 (6.7%)	
PFS (months, CI)	10.0 (8.51–11.55)	10.3 (7.7–12.8)	6.3 (2.0–10.6)	*p* = 0.21
Rate at 6 months	72.0%	75,8%	55.8%	
Rate at 1 year	40.8%	44.7%	23.9%	
Rate at 2 years	26.3%	28.8%	16.0%	
Rate at 3 years	14.3%	14.2%	16.0%	
Rate at 5 years	10.7%	9.5%	16.0%	
OS (months, CI)	36.3 (20.9–51.6)	37.2 (28.0–46.4)	14.0 (1.0–27.1)	***p* = 0.039**
Rate at 6 months	85.8%	91.2%	62.3%	
Rate at 1 year	77.5%	80.9%	62.3%	
Rate at 2 years	56.9%	61.6%	39.0%	
Rate at 3 years	50.7%	57.7%	19.5%	
Rate at 5 years	32.4%	34.7%	19.5%	

## Data Availability

Data available on request due to privacy and ethical restrictions.

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
