# Peer review of "Retrospective Analysis of the Correlation of MSI-h/dMMR Status and Response to Therapy for Endometrial Cancer: RAME Study, a Multicenter Experience"

_cancers, 2023, doi:10.3390/cancers15143639_

Round 1

Reviewer 1 Report

This study aimed to identify therapeutic response of MSI-H/dMMR status in endometrial cancer. For this, the authors divided patients into MMRd and MMRp groups.

1.     If the purpose is to assess the response to therapy, it may be more relevant to examine the association between MMR status and RFS/ PFS, rather than OS. If that is the case, the conclusion should address the association between MMR status and RFS/PFS. Currently, the conclusion in the abstract only focuses on OS, which seems incongruous. If the objective is to investigate the association with OS, the research objective should be modified to reflect the prognostic value.

2.     Currently, molecular subtyping of endometrial cancer is well-established, and it is known that MMR-proficient group includes p53-mutant, POLE, and NSMP subtypes, which have been reported to exhibit different prognoses compared to the MMRd subtype. The results of this study would vary depending on the proportion of these subtypes within the MMR-proficient group. Therefore, further defining the MMR-proficient group is necessary. At the very least, confirming the p53-mutant group and conducting a reanalysis would enhance the significance of this study.

3.     It is necessary to provide more specific details about the treatments received. Currently, it is stated that mainly platinum-based chemotherapy was administered, but it is unclear if other targeted therapies were included or not.

4.     The paragraphs are divided into too many sections. Similar-context paragraphs should be combined.

5.     The K-M graph should include p-values. It would be appropriate to include a legend indicating the colors within the figure.

6.     It is mentioned that MMR IHC or MSI PCR tests were performed for analysis, but it is important to specify the proportion of patients who underwent both tests. If there were patients who underwent both tests, it should be indicated what percentage they comprised and whether the results were concordant.

Minor

1.     The country is missing in Affiliation 1.

Author Response

Reviewer 1  

  1. If the purpose is to assess the response to therapy, it may be more relevant to examine the association between MMR status and RFS/ PFS, rather than OS. If that is the case, the conclusion should address the association between MMR status and RFS/PFS. Currently, the conclusion in the abstract only focuses on OS, which seems incongruous. If the objective is to investigate the association with OS, the research objective should be modified to reflect the prognostic value.

Thank you for your valuable suggestions. We have modified conclusions adding a comment on why RFS advantage does not translate in OS advantage in the dMMR/MSI-h population. In detail dMMR population was undetreated because ICIs were not used.

  1. Currently, molecular subtyping of endometrial cancer is well-established, and it is known that MMR-proficient group includes p53-mutant, POLE, and NSMP subtypes, which have been reported to exhibit different prognoses compared to the MMRd subtype. The results of this study would vary depending on the proportion of these subtypes within the MMR-proficient group. Therefore, further defining the MMR-proficient group is necessary. At the very least, confirming the p53-mutant group and conducting a reanalysis would enhance the significance of this study.

Thank you. Unfortunately, in our database we do not have data about p53 and we are not able to perform this analysis.

  1. It is necessary to provide more specific details about the treatments received. Currently, it is stated that mainly platinum-based chemotherapy was administered, but it is unclear if other targeted therapies were included or not.

Thank you. In our database, for the 77 patients that received systemic treatment for metastatic/advanced disease, we collected information about treatments received as follow: platinum based chemotherapy, doxorubicine lyposomial, other therapy. The results are reported in the table below

MSI-l/pMMR (N=62)

MSI-h/dMMR (N=15)

Platinum-based chemotherapy

46 (74%)

13 (87%)

Doxorubicine-lyposomial

11 (18%)

1 (6.5%)

Other therapy

5 (8%)

1 (6.5%)

The majority of the patients received platinum-based therapy in both groups (74% and 87%) instead of a minority that received doxorucine lyposomial (18% and 6.5%). Due to the small number of patients not receiving platinum-based therapy, we focused our analyses for these patients.  If you think the table adds significant information to manuscript we can upload as supplementary materials.

  1. The paragraphs are divided into too many sections. Similar-context paragraphs should be combined.

Thank you.

We reduced the number of sections in the Methods according to your suggestion.

  1. The K-M graph should include p-values. It would be appropriate to include a legend indicating the colors within the figure.

Thank you.

All the legends contain this statement ‘Blue line MSI-l/pMMR, green line MSI-h/dMMR’. We also added p-values.

  1. It is mentioned that MMR IHC or MSI PCR tests were performed for analysis, but it is important to specify the proportion of patients who underwent both tests. If there were patients who underwent both tests, it should be indicated what percentage they comprised and whether the results were concordant.

Thank you.

The MMR/MSI status was assessed locally, according to one of the standard practices. Methods for determination included loss of 1 or more MMR protein expression by immunochemistry (IHC), or instability in 2 or more of five tumor repeat loci by polymerase chain reaction (PCR) assay. When IHC was used, all the four MMR proteins were assessed (MLH1, PMS2, MSH2, MSH6).

In our series 75% of the patients were tested with IHC, 8% with PCR and 17% with both the methods. When performing both methods of testing, results were concordant.

We added it in the text (See MMR/MSI testing paragraph, line 125-132).

Minor

  1. The country is missing in Affiliation 1.

Thank you. We added Italy in Affiliation 1.

Reviewer 2 Report

The paper is well written, the research is original and the methodology well structured. The authors of the study aimed to identify the impact of MSI-I / dMMR on response of chemotherapy in endometrial cancer. This is an important information for clinical practice. Although no impact was found in the adjuvant setting, a worse survival was seen in the primary chemotherapy group, which indicated better treatment are required for advance endometrial cancer other than chemotherapy i.e immune checkpoint inhibitors. This point should be stressed in the final conclusion.

Further elaboration on the MMR/MSI testing is required (line 125). What kind of IHC for MMR had been performed (MLH1, PMS2, MSH2, MSH6) and had they been done on all patients? How many had IHC/ PCR/ or both?

The introduction included a detailed discussion on immune checkpoint inhibitor which is not the main aim of the study. This can be shortened.

The manuscript is well written except for some typos in the tables.

Author Response

Reviewer 2

  1. The paper is well written, the research is original and the methodology well structured. The authors of the study aimed to identify the impact of MSI-I / dMMR on response of chemotherapy in endometrial cancer. This is an important information for clinical practice. Although no impact was found in the adjuvant setting, a worse survival was seen in the primary chemotherapy group, which indicated better treatment are required for advance endometrial cancer other than chemotherapy i.e immune checkpoint inhibitors. This point should be stressed in the final conclusion.

Thank you.

We stressed in the final conclusion that our study further confirms the need to incorporate ICIs in the MSI-h/dMMR EC population in line with the data from randomized phase III clinical trials.

  1. Further elaboration on the MMR/MSI testing is required (line 125). What kind of IHC for MMR had been performed (MLH1, PMS2, MSH2, MSH6) and had they been done on all patients? How many had IHC/ PCR/ or both?

Thank you.

The MMR/MSI status was assessed locally, according to one of the standard practices. Methods for determination included loss of 1 or more MMR protein expression by immunochemistry (IHC), or instability in 2 or more of five tumor repeat loci by polymerase chain reaction (PCR) assay. When IHC was used, all the four MMR proteins were assessed (MLH1, PMS2, MSH2, MSH6).

In our series 75% of the patients were tested with IHC, 8% with PCR and 17% with both the methods. When performing both methods of testing, results were concordant.

We added it in the text (See MMR/MSI testing paragraph, line 125-132).

  1. The introduction included a detailed discussion on immune checkpoint inhibitor which is not the main aim of the study. This can be shortened.

Thank you the suggestion. However, the editor asked us to extend the main text to reach at least 3000 words.

Round 2

Reviewer 1 Report

  1. The K-M graph should include p-values. It would be appropriate to include a legend indicating the colors within the figure. 

The above suggestion was not addressed in the revised manuscript. The K-M figures need to be revised.